**communications** engineering

# Multidisciplinary approaches in electronic nicotine delivery systems pulmonary toxicology: emergence of living and non-living bioinspired engineered systems
Kambez H. Benam [1,2,3] ✉

Technology-based platforms offer crucial support for regulatory agencies in overseeing tobacco products to enhance public health protection. The use of electronic nicotine delivery systems (ENDS), such as electronic cigarettes, has surged exponentially over the past decade. However, the understanding of the impact of ENDS on lung health remains incomplete due to scarcity of physiologically relevant technologies for evaluating their toxicity. This review examines the societal and public health impacts of ENDS, prevalent preclinical approaches in pulmonary space, and the application of emerging Organ-on-Chip technologies and bioinspired robotics for assessing ENDS respiratory toxicity. It highlights challenges in ENDS inhalation toxicology and the value of multidisciplinary bioengineering approaches for generating reliable, human-relevant regulatory data at an accelerated pace.

The rapid development and proliferation of electronic nicotine delivery systems (ENDS) over the past two decades have considerably impacted society and health, both in the United States and globally. ENDS encompass a wide variety of products, including vapes, vaporizers, vape pens, hookah pens, electronic cigarettes (ECs), e-cigars, and e-pipes. The widespread adoption of these devices is driven by their accessibility and the customizable nature of e-liquids. Among these products, ECs have become the most widely used, which particularly concerning because they appeal to youth and young adults who have never smoked traditional cigarettes.

A key driver behind the rapid adoption of ENDS is the perception that they are a safer alternative to traditional combustible cigarettes[1]. This perception is largely fueled by marketing strategies that emphasize the absence of tar and other harmful combustion by-products; however, ENDS are not harmless[2–5]. The ability to customize e-liquids with various flavors and nicotine concentrations has made ENDS attractive to younger populations. Flavors such as fruit, candy, dessert, mint, and menthol are especially appealing to adolescents and young adults[6], contributing to the initiation of vaping among these groups.

The societal impact of ENDS is multifaceted. There is potential for harm reduction among adult smokers who switch entirely from combustible cigarettes to ENDS, potentially lowering their exposure to harmful chemicals. However, the widespread use of ENDS among non-smoking youth and young adults poses substantial public health challenges. The increasing prevalence of nicotine addiction in these demographics is notably alarming, as it can lead to long-term health consequences and an increased likelihood of transitioning to traditional cigarette smoking. Furthermore, the accessibility of ENDS products through online and retail stores as well as social sourcing has facilitated their adoption[7–10]. Despite regulatory efforts to restrict sales to minors, many young users still find ways to purchase these products. This accessibility, combined with aggressive marketing and social media influence, has considerably contributed to the normalization of vaping in youth culture[2,5,8,10].

## Context and challenges
### Epidemiological data and trends
Recent data indicate that ECs were the most commonly used tobacco product among middle and high school students in the U.S. in 2022 and 2023[6]. In 2023 alone, 22.2% of this population reported ever using any tobacco product[6]. Similarly, a cross-sectional study involving more than 414,000 participants surveyed via the 2021 Behavioral Risk Factor Surveillance System (BRFSS) in the U.S. revealed a vaping incidence of 18.3% among young adults aged 18–24 years[11]. Among those reporting current EC use, more than 20% indicated no prior use of combustible cigarettes[11]. These epidemiological findings underscore the extensive and unintended impact

[1]Division of Pulmonary, Allergy, Critical Care, and Sleep Medicine, Department of Medicine, University of Pittsburgh, Pittsburgh, PA, USA. [2]Department of Bioengineering, University of Pittsburgh, Pittsburgh, PA, USA. [3]Vascular Medicine Institute, University of Pittsburgh, Pittsburgh, PA, USA.
✉e-mail: kambez.benam@pitt.edu

of the ENDS. The popularity of vaping devices especially among adolescents increases the risk of nicotine addiction and potential long-term health consequences. This reach of ENDS is beyond its potential as a less harmful alternative to traditional cigarettes, highlighting the need for targeted regulation and public health interventions to mitigate adverse effects.

## Pulmonary and cardiovascular health impacts

Emerging research has revealed a range of negative health effects associated with the use of ENDS. An analysis of surveys and electronic health records from over 175,000 participants from *All of Us* Research Program—a large national study led by the U.S. National Institutes of Health, Bene-Alhasan et al. revealed that individuals who used ECs were 19% more likely to develop heart failure than those who had never used ECs[12]. Similarly, Mohammadi and colleagues demonstrated that chronic EC use significantly impairs vascular function[13]. Their findings showed that chronic EC users exhibited lower brachial artery flow-mediated dilation than nonusers[13]. Additionally, compared with serum from nonusers, serum from EC users has been shown to reduce vascular endothelial growth factor-stimulated nitric oxide secretion from endothelial cells[13]. The study also identified a significant increase in the circulating levels of endothelial cell adhesion molecules and inflammatory markers, including S100 calcium binding protein A8, high mobility group box 1, interferon β, soluble intracellular adhesion molecule 1, von Willebrand factor, and myeloperoxidase, in EC users[13]. These findings highlight the harmful effects of chronic EC use on vascular endothelial function.

In pulmonary medicine, the use of ECs has been linked to an increased incidence of respiratory symptoms in adolescents and young adults[14–16]. Tackett et al. evaluated a sample of over 1500 U.S. high school students, with a mean age of 17.3 years, over four separate occasions (2014, 2015, 2017, and 2018). The study found that past 30-day EC use increased the odds of wheezing (odds ratio [OR] 1.41), bronchitic symptoms (OR 1.55), and shortness of breath (OR 1.48) after adjusting for various factors such as study timepoint, age, sex, race, lifetime asthma diagnosis, parental education, current cigarette use, cannabis use, and secondhand exposure to ECs, cigarettes, and cannabis[14]. McConnell and colleagues reported similar results in their analysis of 2,086 participants from the Southern California Children's Health Study. They found that the risk of bronchitic symptoms was greater among past (OR 1.71) and current (OR 1.41) EC users, based on the questionnaires completed by 11th and 12th-grade students[15]. Out of the total sample, 502 participants had ever used ECs, with 301 being past users and 201 being current users[15]. Internationally, Alnajem et al. conducted a cross-sectional study involving 1345 high school students aged 16–19 years in Kuwait. The study revealed that 27.4% of the participants reported EC use[16]. Compared with nonusers, those with no history of cigarette smoking had an increased prevalence of wheezing (adjusted prevalence ratio [PR] 1.54) and asthma symptoms (adjusted PR 1.85)[16]. These studies collectively highlight important respiratory risks associated with EC use among adolescents and young adults, emphasizing the need for increased awareness and regulatory measures to mitigate these health impacts.

## Chemical composition and toxicity of e-liquids

E-liquids, also known as vape liquids, are liquids within an ENDS that, when heated, transform into inhalable aerosols and deliver nicotine, flavorings, and/or other ingredients to vaping device users without the combustion process associated with traditional tobacco products. The primary constituents of any e-liquid are propylene glycol (PG) and vegetable glycerin (VG). PG and VG are also referred to as EC humectants, solvent carriers, or vaporizing solvents; they lead to aerosol production, which simulates combustible tobacco cigarette smoke (CS)[17].

Multiple studies have revealed the toxicity of PG and VG[18–23]. PG, which when heated, accounts for ~30–95% of an EC liquid by volume[24–27], can generate toxic respiratory chemicals, including diacetyl, formaldehyde, and methylglyoxal[18,19]. Komura et al. reported that exposure of primary healthy and chronic obstructive pulmonary disease (COPD)-derived human small airway epithelial cells to 4% PG significantly inhibited cell proliferation and viability, induced apoptosis, led to DNA damage, and caused cell cycle arrest in vitro[20]. Similarly, PG:VG negatively impacts the viability of human airway smooth muscle cells[21]. In another report, Woodall and colleagues showed that treating primary human bronchial epithelial cells with a PG or PG:VG mixture for as little as 30 min inhibited glucose transport and metabolism, reduced cell height, lowered barrier function, and increased airway surface liquid height[22].

## Case studies: EVALI and diacetyl exposure

The findings discussed above clearly indicate the toxic potential of e-liquid components and their vaporized aerosols. However, given the vast diversity of flavorings, nicotine concentrations and types, and other chemicals (many of which are proprietary) that can be added to PG:VG mixtures, along with the variety of PG:VG ratios that can be prepared, numerous e-liquids can potentially be obtained. This creates a new layer of complexity and uncertainty when predicting the potential toxicity of any given e-liquid. Muthumalage et al. showed that the treatment of human monocytic cell lines with flavoring chemicals and nicotine-free flavored e-liquids caused cytotoxicity and induced secretion of the pro-inflammatory chemokine IL8 in vitro[28]. In 2014 alone, more than 7000 unique e-liquid flavorings were commercially available online[29], and the range of flavorings has grown considerably since then. Therefore, there is an urgent need for rapid evaluation of emerging ENDS products to determine their potential pulmonary toxicity.

This need is exemplified by instances in which seemingly safe compounds are mixed and matched in e-liquids by users, but the exposure to aerosols of such ENDS has led to unexpectedly high morbidity and/or mortality. For instance, between 2019 and 2020, the United States reported 2807 cases of e-cigarette, or vaping, product use-associated lung injury (EVALI), which led to 68 deaths as of October 2020, representing ~2.4% mortality rate[30,31]. Initial analyses of bronchoalveolar lavage fluid (BALF) identified vitamin E acetate (VEA) in 48 out of 51 patients with EVALI across 16 states[32]. Following a nationwide investigation, the U.S. Centers for Disease Control and Prevention (CDC) announced that the addition of VEA to e-liquids is strongly linked with the EVALI epidemic[33]. Systematic studies revealed that the majority of EVALI patients required hospitalization, with around 95% needing inpatient care, and ~27% of those admitted requiring intubation[31]. Additionally, mortality rates were higher among older patients and those with pre-existing conditions such as cardiac disease, asthma, or mental health issues[34,35]. Moreover, a 12-month follow-up of EVALI survivors showed that 39% of patients had cognitive impairment, whereas 48% experienced respiratory limitations[36].

Another example of a chemical added to vaping liquids that was initially considered safe is diacetyl, also known as 2,3-butanedione. This substance is a member of a class of organic compounds called diketones and is known for imparting a buttery flavor[37]. Diacetyl occurs naturally in some foods and is also used as a synthetic flavoring agent in products such as butter, cocoa, coffee, and dairy products[37]. While it has been approved for ingestion and considered safe by the National Institute for Occupational Safety and Health (NIOSH)[38] and the Flavor and Extract Manufacturers Association (FEMA) under FEMA GRAS No. 2370, inhalation of diacetyl is a different matter. Workers exposed to diacetyl-containing flavorings have reported an excessive decline in forced expiratory volume in one second (FEV1), which indicates respiratory impairment[39]. Additionally, diacetyl has been linked to higher risk of developing bronchiolitis obliterans, COPD, and other severe pulmonary conditions[40–42]. This condition was first identified in workers at popcorn factories who were exposed to elevated levels of diacetyl used in artificial butter flavoring. Unfortunately, despite its harmful respiratory effects, when inhaled, several studies have shown that diacetyl is one of the most prevalent chemicals in EC vapors[37,43,44].

## Global regulatory disparities and challenges

Global disparities in the regulation and public perception of ENDS add further complexities to the increasing popularity and utilization of these products. Divergent approaches, ranging from stringent regulations in

certain countries to more permissive attitudes elsewhere, underscore the imperative for international cooperation and standardized guidelines to address the multifaceted challenges posed by ENDS effectively. For instance, the World Health Organization recently reported substantial discrepancies in regulatory frameworks across nations. As of July 2023, shockingly, 88 countries lacked a minimum age requirement for the purchase of vaping devices, while an alarming 74 countries, predominantly African countries alongside Mongolia, Pakistan, and Colombia, had no established regulations safeguarding public health against these potentially harmful products[45]. In contrast, 34 countries, including Iran, India, Thailand, and Brazil, implemented outright bans on ENDS[45]. These disparities highlight the need for concerted global efforts to establish cohesive regulatory frameworks that prioritize public health and safety. Harmonizing regulations and fostering collaboration among nations can help mitigate the adverse effects of ENDS and promote more consistent and effective measures to address their proliferation and use worldwide.

As discussed above, the widespread use of ENDS has led to substantial societal and health implications. The varied ENDS products, including vapes and ECs, appeal to a broad demographic, including youth, raising public health concerns. The negative health effects, such as cardiovascular and respiratory issues, underscore the urgent need for regulatory frameworks. The EVALI epidemic and the toxicity of compounds such as VEA and diacetyl highlight the severe risks associated with ENDS. Given these complexities, it is crucial to utilize advanced and human-relevant preclinical models to study the inhalation toxicology of ENDS comprehensively. Figure 1 and Table 1 provide an overview of the diverse models discussed in this article. These models offer varying degrees of physiological relevance and complexity, helping researchers understand the multifaceted effects of ENDS exposure and developing strategies for public health protection.

## Existing preclinical living models for the pulmonary toxicological assessment of ENDS

The impact of vaping has been predominantly studied via models of the lung due to inhaled exposure to EC vapors. These studies have provided meaningful insights into the respiratory effects of vaping, including inflammation, impaired lung function, and increased risk of respiratory diseases. However, there are also preclinical and clinical reports that have examined the effects of vaping on other organ systems, indicating potential broader systemic health risks[12,13,46–53]. Despite these important findings, the focus of this paper is confined to the pulmonary impacts of vaping.

A critical aspect of addressing the societal and health impacts of emerging ENDS and regulatory needs is the development of new preclinical tools. These tools should be human-centric, physiologically and clinically relevant, and capable of rapidly generating data for toxicological assessments. Traditional methods often struggle to keep pace with the dynamic landscape of emerging products, necessitating innovative research models that accurately simulate human responses to ENDS exposure.

### 2D and 3D static cell culture models

Several two-dimensional (2D) and three-dimensional (3D) static cell culture models have been applied to evaluate the pulmonary toxicity of ENDS. These models often involve human lung epithelial cell lines or primary cells cultured either on traditional plates or on the porous membranes of transwell inserts (TWIs)[23,54–56]. For instance, Anthérieu et al. utilized the human bronchial epithelial cell line Beas-2B at an air-liquid interface (ALI) to compare the toxicity of EC vapors against CS[54]. Similarly, Sinha and colleagues used Beas-2B cells to investigate the cytotoxicity and inflammatory response to EC aerosols[55]. While these studies offer valuable insights, due to their specific origins and characteristics, into the toxicological effects of ENDS and can potentially be performed in a medium- to high-throughput manner, they are limited by the virtue of using cell lines. For example, Beas-2B cells, though non-cancerous, have been immortalized by viral infection to express the SV40 T antigen and lack the ability to differentiate well into mucociliated bronchial epithelium[57]. Additionally, these

cells exhibit surface marker expression profiles and osteogenic and adipogenic differentiation potentials similar to those of human umbilical cord-derived mesenchymal stem cells[58].

In contrast, the differentiation of primary human airway epithelial cells (hAEpCs) into ciliated epithelia under ALI provides a histologically more relevant model system for analyzing ENDS-related respiratory toxicity. Manna et al. exposed mucociliated hAEpCs cultured in TWIs to EC vapors and assessed changes in cellular populations and the secretion of immunomodulatory cytokines[56]. The use of differentiated hAEpCs enables researchers to replicate various epithelial subtypes, including ciliated cells, mucin-producing goblet cells, Club cells, and basal cells, mimicking the pseudostratified columnar epithelium of the human airway. However, several limitations affect this approach: (1) inability to replicate the dynamic interaction between the lung epithelium and the systemic immune system, including the recruitment and activation of circulating leukocytes; (2) limited multicellularity, as most models only reconstitute the epithelium in vitro; (3) lack of blood-like fluidic flow, resulting in the absence of physiological vascular shear and dynamic endothelial cell–immune cell interactions in these model systems when epithelium is in co-culture with endothelial cells; and (4) absence of a physiologically relevant and thick sub-epithelial extracellular matrix (ECM). Furthermore, at best, hAEpC ALI cultures are exposed to vertically delivered puffs of EC aerosols under static conditions, whereas human ENDS users typically exhibit characteristic patterns of puff durations, puff volumes, and inter-puff intervals (referred to as vaping topography) *combined with* breathing mechanics, resulting in dynamic exposure to EC aerosols under conditions of inhale-exhale airflow that apply horizontal shear forces across the airway epithelium[59,60].

### Lung spheroids and organoids

Spheroids and organoids derived from lung cells constitute another potential culture system for pulmonary toxicology studies. Spheroids are 3D sphere-like clusters of cells that can self-assemble from single-cell suspensions[61–67]. They are typically derived from tumor cells or immortalized lines (e.g., A549)[62], primary cells such as outgrowth progenitor cells from lung tissue explants or cells obtained from transbronchial lung biopsies[63–65], or induced pluripotent stem cells (iPSCs)[66,67]. Lung spheroids can mimic the alveoli and airway epithelium. Organoids, on the other hand, are more complex 3D structures that arise from stem or progenitor cells, and are capable of self-organizing into miniaturized and simplified versions of organs such as alveolar-like sacs and bronchiole-like, airway structures[61,68,69]. Compared with spheroids, lung organoids exhibit self-renewal capacity and consist of more than one cell type or tissue. Additionally, they exhibit long-term viability and depend on intrinsic developmental processes rather than solely on cell-cell adhesions to form tissue- and organ-like microarchitectures[61].

The application of lung organoids and spheroids to study the toxicity of ENDS has been very limited. As of July 2024, only one study has utilized bronchial spheroids derived from Beas-2B cells to investigate the effects of diesel exhaust particles[70]. This study revealed that exposure to diesel exhaust particles significantly impacts the viability and function of bronchial spheroids, leading to increased production of pro-inflammatory cytokines and oxidative stress markers.

Despite their potential, there are limitations to organoids and spheroids. Spheroids, while valuable, lack the complexity of full organ systems and may not fully recapitulate the lung microenvironment. They also lack vasculature and ALI, which are essential features for accurately simulating lung physiology. Furthermore, controlling cell ratios and aggregate sizes in spheroid cultures can be challenging, leading to variability in experimental outcomes. Organoids, though more complex, can exhibit variability in size and cellular composition, and their generation and maintenance require specialized techniques and expertise. They lack several key organ features such as vascularization, and it can be challenging to obtain fully differentiated lung cell types. Importantly, neither model can fully replicate the dynamic mechanical forces with blood and air flows experienced in vivo.

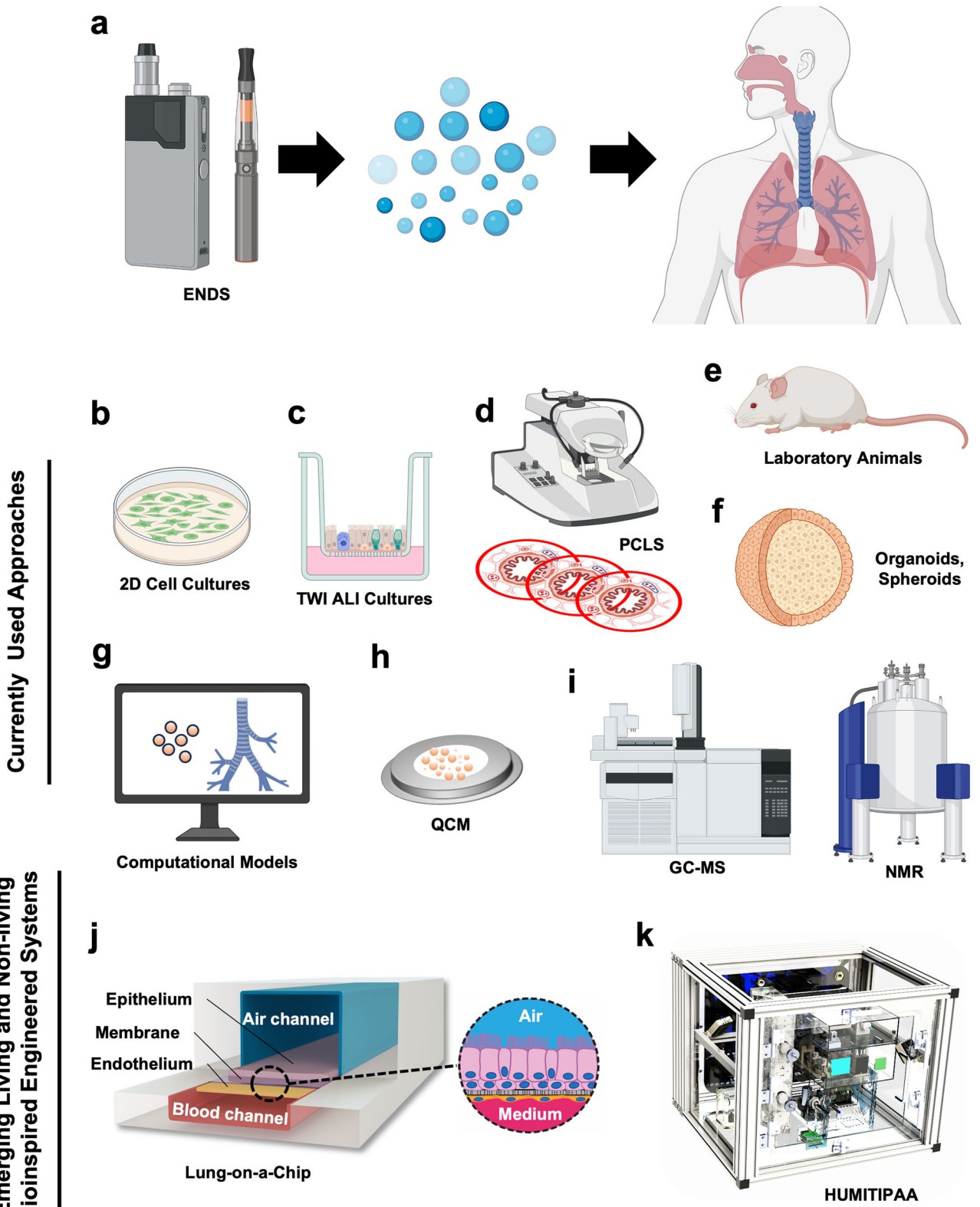

## Precision-cut lung slices

Precision-cut lung slices (PCLS)[71–74], uniform sections of the lung measuring 150–500 μm in thickness, have emerged as an organotypic ex vivo model that addresses the limitations of cell lines and primary human lung epithelial cell ALI cultures. In the context of ENDS toxicology, few studies have been performed using PCLS. Crue et al. exposed human PCLS from a single donor to vaping extract with or without influenza A virus (IAV) infection, followed by single-cell RNA sequencing to analyze changes in host antiviral and pro-inflammatory responses[75]. The vaping extract was prepared by passing vapors from a commercially available EC through tissue culture medium. Similarly, the same research group treated PCLS obtained from the lungs of five non-smoking healthy human donors with e-liquid and IAV to investigate the viral

**Fig. 1 | Evolving Landscape of Preclinical Tools Used for Toxicological Assessment of Electronic Nicotine Delivery Systems. a** Electronic nicotine delivery systems (ENDS) are proliferating swiftly, with a wide array of e-liquid ingredients and device functionalities, presenting considerable challenges in assessing their potential pulmonary toxicity. Traditional methodologies predominantly include two-dimensional (2D) cell culture models that employ either cell lines or undifferentiated primary human lung cells (**b**), as well as static three-dimensional (3D) in vitro models such as air-liquid interface (ALI) cultures using primary human airway epithelial cells in transwell inserts (TWIs) (**c**), microtome-assisted precision cut lung slices (PCLS) (**d**), laboratory animals, notably rodents (**e**), and, rarely, lung spheroids and organoids (**f**). Complementary techniques involve in silico modeling (**g**) and physicochemical analytical methods such as quartz crystal microbalance (QCM), gas chromatography-mass spectrometry (GC-MS), and nuclear magnetic resonance (NMR) spectroscopy (**h, i**). Emerging bioengineered technologies, such as Lung-on-a-Chip living devices (**j**) and the bioinspired non-living robotic platform Human Vaping Mimetic Real-Time Particle Analyzer (HUMITIPAA) (**k**), offer new opportunities to overcome certain limitations of existing tools, enhance human and clinical relevance, and expedite toxicity evaluation. Images in (**j, k**) were reproduced from ref. 87. and ref. 138, respectively. **a–g, i** Were fully or in part created using BioRender.com, and (**j, k**) were reproduced with permission from ref. 87 and ref. 138.

load and immune response[76]. PCLS obtained from animals, such as chicks (Gallus gallus)[77] and laboratory mice (strain C57BL/6)[78], have also been utilized for evaluating EC toxicity. Despite their ability to closely mimic the cellular and architectural complexity of the (human or animal) lung, PCLS has several drawbacks. These limitations include relatively low viability in culture; typically necessitating experimentation within 5–10 days of slicing; inability to recreate physiological ALI and breathing and exposure to ENDS aerosols under rhythmic inhalation airflow; lack of vascular circulation; inability to recreate dynamic intercellular interactions between circulating immune cells and vascular endothelial cells; considerable heterogeneity in cellular composition and responses to stimulation between slices obtained from different locations of the same donor lung or single lobe, as well as across different donors; limited availability of human lungs for slicing; and challenges in standardization and automation owing to manual processing and handling, apart from slicing via a microtome.

## Animal models

To complement the in vitro and ex vivo experimental approaches discussed above, researchers have turned to animal models[79–83]. In this context, laboratory mice and rats have been the most extensively utilized in vivo models[79–81]. For example, Ha et al. conducted a study in which wild-type C57BL/6 mice were exposed to EC aerosols for 4 months and evaluated for inflammatory and immune responses in the larynx and compared them to those in mice exposed to CS[79]. In another study, mice were acutely challenged with aerosols generated from a PG:VG e-liquid supplemented with VEA, and the total number of leukocytes infiltrating the lungs, as well as the levels of VEA and albumin in the BALF, were analyzed[80]. Similarly, Rau and colleagues exposed Sprague Dawley rats to EC vapors for 4 weeks to investigate the toxic microcirculatory effects of exposure[81].

Animal models offer insights into the local (pulmonary) and systemic effects of ENDS aerosols, as well as their impact on other organ systems. However, ethical considerations on the use of animals in biomedical research and interspecies differences limit the extrapolation of findings to humans. For instance, the microscopic structure of the lungs, airway branching, cellular distribution, and immune system functioning differ substantially between rodents and humans[84–87] (e.g., neutrophils constitute 10–25% of circulating leukocytes in mice vs. 50–70% in humans[85,87]). Additionally, the number of non-ciliated Club-like cells in the small airways of rats and mice is ~10-fold greater than that in humans, and goblet cells are almost absent in the membranous bronchioles of mice and rats[86]. Moreover, rodents are obligate nose breathers with intricate and highly developed nasal turbinates that lead to different particulate deposition patterns than humans[88]. Furthermore, replicating human-relevant underlying conditions, such as former smoking status or co-morbidities such as idiopathic pulmonary fibrosis (IPF), COPD airway disease, or severe corticosteroid-resistant asthma, can be difficult, if not impossible. Additionally, observing the emergence of an EC-induced phenotype of interest in animal models can be a time-consuming process, often spanning several weeks to months.

## Organ-on-a-Chip models of inhaled exposure to ENDS aerosols

Organs-on-Chips[87,89–102], commonly referred to as microphysiological systems (MPS), are microfluidic cell culture devices designed to mimic the structure and function of human tissues and organs at a miniature scale. These experimental research tools predominantly consist of a transparent, flexible, and gas-permeable polymer, like polydimethylsiloxane (PDMS), housing hollow microfluidic channels lined with living human cells and/or chambers filled with cell-free or cell-laden hydrogels. By recapitulating the multicellular architectures, tissue–tissue interfaces, chemical gradients, mechanical cues, and vascular perfusion of natural organs, these devices reproduce levels of tissue and organ functionality that are not possible with conventional 2D or 3D culture systems. They also enable high-resolution, real-time imaging and in vitro analysis of the biochemical, genetic, and metabolic activities of living *human* cells in a functional *human* tissue and organ context. Organs-on-Chips offer a platform for studying disease mechanisms, drug toxicity, and efficacy as well as inhalation toxicology in a more physiologically relevant context. Moreover, they can be interconnected to create multi-organ systems, allowing researchers to study organ–organ interactions and the systemic effects of drugs and toxins. As such, Organs-on-Chips hold remarkable potential for advancing biomedical research, including tobacco-related toxicology, and provide an opportunity to address limitations of existing preclinical living lung models of ENDS exposure.

## Lung Small Airway-on-a-Chip

The application of Organs-on-Chips technology in the toxicology of ENDS is still emerging. Benam et al. pioneered this approach by developing an all-primary Human Lung Small Airway-on-a-Chip[87]. This biodevice integrated two channels: an airway channel coated with human-derived mucociliary bronchiolar epithelium maintained at an ALI and a vascular channel that simulates post-capillary venules lined with human microvascular endothelium under fluid flow. These channels were separated by a porous membrane that allowed nutrient exchange. The biodevice enabled the investigators to replicate and analyze lung airway pathophysiology at the organ level in vitro, including modeling disease-relevant human lung airway inflammation, detecting synergistic effects of the lung endothelium and epithelium on cytokine secretion, and quantifying the recruitment of circulating immune cells in response to activation by the viral mimic poly I:C, demonstrating similar trends in biological responses among chips fabricated with cells from the same donor population, identifying a new biomarker of disease exacerbation induced by viral infection, and measuring therapeutic responses to anti-inflammatory drugs that inhibit cytokine-induced recruitment of circulating neutrophils under flow[87].

## Breathing-Smoking/Vaping Human Lung-on-a-Chip

In a follow-up study, the same research team expanded on this model and created a modular and microfluidically integrated platform that allowed in vitro analysis of the effects of whole cigarette smoke and EC aerosols delivered under physiologically relevant flow conditions that mimic breathing on the pathophysiology of differentiated human mucociliated bronchiolar epithelium[91]. This platform, referred to as the Breathing-Smoking/Vaping Human Lung-on-a-Chip, consisted of the Lung Small Airway-on-a-Chip—the living conducting airway tissue to be exposed to inhaled smoke/vapors; a micro-respirator that mimicked ribcage and the diaphragm to reproduce breathing mechanics and patterns; a micro-fluidically compatible fresh whole CS and EC vapor generator; and control

**Table 1 | Preclinical Platforms for ENDS Inhalation Toxicology. Various systems, living and non-living, bioengineered and non-bioengineered, reviewed in this article**

| System Classification | Components | Characteristics (Pros and Cons) | Provided Biological Information | Application Examples |
|---|---|---|---|---|
| 2D and 3D Static Cell Culture Models | Human lung epithelial cell lines (e.g., Beas-2B); TWI models of primary hAEpCs | **Pros:** 2D Cultures: easy to culture, amenable for medium-to-high throughput testing. TWIs: recreation of in vivo-seen differentiated airway epithelium. **Cons:** 2D Cultures: lack differentiation into mucociliated airway epithelium, do not capture inter-individual variability. TWIs: lack dynamic interaction between lung epithelium and systemic immune system, limited multicellularity, absence of blood-like fluidic flow and physiological vascular shear, lack sub-epithelial extracellular matrix. | Cytotoxicity, inflammatory response, oxidative stress. | BEAS-2B cells to investigate cytotoxicity and inflammatory response to EC aerosols. |
| Lung Spheroids and Organoids | 3D structures mimicking airway and alveoli derived from lung cells, iPSCs or tumor cells | **Pros:** More closely mimic tissue architecture and cellular interactions. **Cons:** Lack vasculature, ALI, hard to control cell ratio and aggregate size, inability to recapitulate dynamic mechanical forces with blood and air flows. | Differentiation into various lung cell types, cellular interactions. | Bronchial spheroids (derived from BEAS-2B cells) to study response to diesel exhaust particles. |
| Precision Cut Lung Slices (PCLS) | Uniform whole lung sections (150–500 µm thick) | **Pros:** Mimic cellular and architectural complexity of lung. **Cons:** Limited viability, lack vascular circulation, high variability, absence of epithelial ALI. | Tissue-level responses, antiviral responses. | Human PCLS to study effects of vaping extract and IAV infection. |
| Animal Models | Mostly mice (e.g., C57BL/6) or rats (e.g., Sprague Dawley) | **Pros:** Systemic effects, multi-organ responses. **Cons:** Ethical concerns, interspecies differences, time-consuming. | Pulmonary and systemic inflammation, immune responses, toxicological outcomes. | Mice exposed to EC aerosols to study inflammatory and immune responses. |
| Organs-on-Chips | Microfluidic devices populated with human cells (e.g., Lung Small Airway-on-a-Chip, Vasculature-on-a-Chip) | **Pros:** Reproduce tissue–tissue interfaces, mechanical cues, vascular perfusion. **Cons:** Low throughput, costly, lack certain tissue complexities. | Lung pathophysiology, cytokine secretion, immune cell recruitment, therapeutic responses. | Human Lung Small Airway-on-a-Chip to study response to EC aerosols under physiologically relevant breathing conditions. |
| Computational Approaches | QSAR models, CFPD-PBTK models | **Pros:** Predict toxicity and particle deposition, reduce animal usage. **Cons:** Require high-quality data, need validation, may not account for all physical and chemical interactions. | Predictive toxicology, pharmacokinetics, molecular interactions. | QSAR models to predict inhalation toxicity of ENDS chemicals. |
| Physico-chemical Analyses | QCM, Cascade Impactors, NMR, GC-MS | **Pros:** Provide detailed chemical composition and physical characteristics. **Cons:** Often terminal endpoints, not real-time, scalability issues. | Particle size distribution, chemical composition, toxic compound identification. | QCM to quantify particulate mass from aerosols, GC-MS for chemical analysis of e-liquids. |
| Bioinspired Robotics | HUMITIPAA | **Pros:** Simulate human vaping behavior and respiration mechanics, measure real-time particle inhalation. **Cons:** Need expansion for measuring nanoparticles, association with biological impact needed. | Real-time particle size distribution, inhaled particle quantity, inhalation dynamics. | HUMITIPAA to assess inhaled particle profiles (quantity and size distribution) from ECs containing VEA or menthol. |

Column titles are noted in bold.

*ENDS:* electronic nicotine delivery systems, *TWI* transwell insert, *ALI* air-liquid interface, *EC* electronic cigarette, *hAEpCs* human airway epithelial cells, *2D* two-dimensional, *3D* three-dimensional, *iPSCs* induced pluripotent stem cells, *PCLS* precision-cut lung slices, *QSAR* quantitative structure-activity relationship, *CFPD-PBTK* computational fluid-particle dynamics-physiologically based toxicokinetic numerical model, *QCM* quartz crystal microbalance, *NMR* nuclear magnetic resonance spectroscopy, *GC-MS* gas chromatography-mass spectrometry, *HUMITIPAA* Human Vaping Mimetic Real-Time Particle Analyzer, *VEA* vitamin E acetate.

software that represented the brain of a human smoker/vaping device user to recreate smoking/vaping topography parameters[91,96]. The motive for developing such a system was that smoking represents a major risk factor for COPD and is a leading non-infectious cause of exacerbations; however, characterizing smoke-induced injury responses under physiological breathing conditions in humans is challenging due to patient-to-patient variability. By applying multi-omics analyses, including gene microarrays, this work identified novel biomarkers for COPD patients who might exhibit greater sensitivity to smoke and novel therapeutic target candidates. In addition, the authors used this platform to expose mucociliated bronchiolar epithelia to aerosols of a first-generation disposable EC and studied the induction of oxidative stress and changes in ciliary beating frequency[91].

### Vasculature-on-a-Chip

In another study, Ohashi and colleagues used a commercially available Vasculature-on-a-Chip model to study the effects of aerosols from heated tobacco products (HTPs) on macrophage adhesion, endothelial expression of the surface intracellular cell adhesion molecule 1, and secretion of pro-inflammatory cytokines[103]. The device consisted of two channels, one of which was used for seeding primary human coronary artery endothelial cells and the other for loading a porcine collagen I-based hydrogel. THP-1 cells were used as macrophages in this study.

### Applications beyond inhalation toxicology

Notably, MPS have been applied successfully in several other fields. In drug development and safety testing, Liver-on-a-Chip models predict human hepatotoxicity and detect cross-species toxicological risks[104,105]. In cancer research, Tumor-on-a-Chip models replicate tumor microenvironments, aiding in the discovery of anti-cancer drugs[106–110]. Neurological diseases have benefited from Brain-on-a-Chip and Blood Brain Barrier-on-a-Chip models, facilitating studies on neurodegenerative diseases like Alzheimer's and Parkinson's, leading to new drug targets and therapeutic candidates[111–116]. Additionally, cardiovascular research has utilized Heart-on-a-Chip models to study cardiac physiology, identify cardiotoxic compounds, and ensure safer therapeutic options[117–120].

### Challenges and future directions

The human Lung Small Airway-on-a-Chip[87] overcomes several limitations of the most advanced lung airway preclinical models that are currently used, namely, TWI ALI culture and PCLS. These include the reproduction of multicellular co-culture of mucociliated hAEpCs with lung microvascular endothelial cells; the introduction of freshly generated ENDS aerosols under rhythmic breathing airflow without disrupting ALI or the need to deliver vapors vertical to the plane of cells; the recreation of in vivo-present mechanical forces, such as shear stress from vascular flow over endothelial cells; the recapitulation of dynamic intercellular interactions between circulating immune cells and the endothelium; and the ability to maintain in vitro viability and functionality of cells on-chip for several weeks. However, this technology is not without its challenges. It is considered low-throughput and can be costly to establish initially, generally requiring specialized expertise for utilization by those not trained in bioengineering. Consequently, generating data on the toxico-pathology of ENDS aerosols is not as expedient or feasible as might be desired. Furthermore, the model currently lacks certain complexities in lung tissue, such as the sub-epithelial ECM and matrix-embedded stromal cells, which are essential for fully characterizing biological responses. Moreover, physiologically relevant geometries (e.g., cross-sectionally circular microvascular lumens), rather than cross-sectionally rectangular or square-shaped blood vessels, need to be reproduced so that the shear forces of flow are uniformly distrusted across the endothelial barrier[99,121]. These represent the next frontiers for advancement as the platform continues to evolve.

The Vasculature-on-a-Chip developed by Ohashi et al., while a first step in delineating endothelium-immune cell interactions in the vascular network, has drawbacks that need to be addressed in future studies. For instance, THP-1 cells are a human acute monocytic leukemia cell line and may not accurately reproduce the biology and functioning of either monocytes or macrophages. The utilization of primary immune cells would be more relevant. In addition, multiple donors need to be investigated (from any given population of interest) to map our inter-individual variability. Moreover, it would me physiologically more relevant if in vivo-seen intercellular interactions are simulated. Macrophages are considered innate immune cells that normally reside in peripheral tissues and lymphoid organs[122] rather than circulating in the peripheral blood. Thus, the evaluation of innate immune cells such as monocytes and granulocytes with the endothelium would be more relevant. Enhancing the ability of the device to recreate a rounded vascular channel would further emulate in vivo-like features.

## Computational and physicochemical analyses of ENDS liquids and aerosols

To better understand the potential health risks associated with ENDS exposure, researchers have also applied in silico modeling alongside physical or chemical characterization of e-liquids and ENDS aerosols[123–136]. These complementary approaches have provided insights into the composition, behavior, and potential toxicological effects of ENDS products.

### Computational fluid-particle dynamics models

Haghnegahdar et al. developed a new multiscale computational fluid-particle dynamics (CFPD)-physiologically based toxicokinetic numerical model to predict local pulmonary deposition and systemic absorption of nicotine and acrolein present in EC aerosols[123]. This model aimed to elucidate how variations in puff topography could influence the deposition and translocation of these molecules[123]. Similarly, Zhao et al. employed a CFPD–Pharmacokinetics (PK) model to quantify the deposition of Δ9-tetrahydrocannabinol (THC) particles and vapor absorption in the human upper airways[124]. Through this computational approach, the authors identified anatomical features and vaping topographic parameters that noticeably impact THC absorption and PK behavior[124]. While these studies[123,124] helped us better understand the effects of anatomical variability on particle deposition and the PK of inhaled cannabis, they lacked explicit benchmarking with human datasets or known toxic compounds. As such, the predictive accuracy of these models has not been fully demonstrated.

### In silico toxicity prediction

Other in silico methods have been utilized to predict the cellular toxicity of ENDS constituents. For instance, Kang et al. conducted structural similarity analysis on flavoring chemicals and byproducts found in tobacco products, including ENDS liquids and aerosols, developing a computational model to predict the genotoxicity and mutagenicity of these chemicals[125]. Similarly, Zarini et al. investigated the applicability of in silico methods for the initial safety assessment of ENDS[126]. The authors evaluated the potential of quantitative structure-activity relationship (QSAR) models to predict the oral and inhalation toxicity of e-liquid constituents. The study involved screening over 200 compounds via QSAR models. They applied LC50 values (lethal concentration for 50% of the population) to assess the inhalation toxicity of e-liquid ingredients. Specifically, the authors evaluated 28 e-liquid ingredients for rats and 23 for mice. Among these, only 22 and 19 compounds served as training datasets for rats and mice, respectively. The inhalation toxicity potential was studied for only 10 compounds—that is test dataset (6 for rats and 4 for mice). While the study showed a predictability level of 86–88%, there were major limitations to the model used. (1) The primary data for model training and benchmarking were derived from rodent (not human) studies. This poses a challenge as toxicological responses in rodents may not accurately reflect those in humans, potentially leading to less reliable predictions of human health risks. (2) The utilized models relied heavily on LC50 data, which classifies doses on the basis of their lethality potential. This approach necessitates the use of high-dose exposures to determine lethal concentrations, which may not be representative of typical human exposures to e-liquid ingredients. Additionally, LC50 data are often limited and may not be available for all relevant compounds, restricting the model's applicability and accuracy.

In silico approaches offer solutions to the limitations associated with acquiring high-resolution local dosimetry data on inhaled ENDS aerosols, allowing estimation of the absorption of individual vapor components under various vaping topography profiles and predicting the potential cellular toxicity of e-liquid ingredients when information is scarce. Moreover, these findings support the reduction in and/or replacement of animal usage in preclinical research. However, computational modeling also has major drawbacks. These challenges mainly revolve around data quality, model validation, and the breadth and depth of the parameters that can be modeled. Validation and clinical correlation are essential for establishing the reliability of in silico methods. However, validating particle deposition across different regions of the lung airways or alveoli or analyzing the absorption of individual ENDS vapor components without taking into account a broad range of intermolecular chemical and physical interactions can be extremely challenging. Computational models often have limited input regarding the number and diversity of anatomical features (e.g., upper or lower airways), despite known inter-individual variability. Additionally, researchers may struggle to predict the toxic effects or absorption of e-liquid constituents beyond the respiratory and vascular systems. Emulating certain factors, such as particle charge, hydrophilicity, concentration, interception, thermophoresis, and gas properties, is inherently difficult[137]. Furthermore, accurately modeling the interactions between inhaled EC aerosol constituents and biological materials such as surfactants remains a complex challenge. Thus, while in silico modeling holds promise for advancing our understanding of ENDS-related health risks, careful validation, and refinement are necessary to ensure the reliability and clinical relevance of the generated data.

### Physical and chemical characterization of ENDS aerosols

Several researchers have adopted physical measurements of inhaled particles as a means to predict the potential toxicity of various ENDS products[127–132]. For instance, Thorne et al. utilized a quartz crystal microbalance (QCM) installed within a smoking machine to quantify the particulate mass deposited from aerosols generated by a cigalike and a closed modular EC[127]. Additionally, the authors eluted nicotine from the QCM surface for further analysis[127]. Similarly, Goros et al. employed QCM to analyze the particle dose of ECs containing eight different flavoring chemicals[128]. In other studies, researchers have utilized cascade impactors for experimental aerosol dosimetry. For example, Oldham et al. determined the particle size distribution from multiple ENDS products using a low-flow cascade impactor[129], a strategy also tested by several other research groups[130–132]. In addition to physical analysis of ENDS aerosols, chemical analytical approaches have been employed to gain insights into the toxicity of emerging vaping products. These methods include nuclear magnetic resonance (NMR) spectroscopy[133] and gas chromatography-mass spectrometry (GC-MS)[134] on e-liquids and thermal desorption-gas chromatography-time of flight mass spectrometry (TD-GC-TOFMS)[135] or ceramic wick[136] analysis on ENDS aerosols to identify harmful or potentially harmful constituents.

Despite their advantages, the physicochemical analyses are often performed as terminal endpoints (rather than in real-time) and in the absence of fresh ENDS aerosol generation while simultaneously mimicking the vaping topography and breathing mechanics. Additionally, scaling up such strategies for medium- to high-throughput analysis may pose feasibility challenges. Furthermore, variability between and within experiments may affect the interpretability of the data.

A number of commercially available particle size-analyzing spectrometers exist, including aerodynamic particle sizers (APS), scanning mobility particle sizers (SMPS), and fast mobility particle sizers (FMPS). APS relies on the principle of inertia to size particles, while the SMPS and FMPS utilize the physical principle of electrical mobility, where a particle's ability to traverse an electric field (its electrical mobility) correlates with its size. SMPS detectors employ condensation particle counters for sensing, whereas FMPS systems utilize multiple low-noise electrometers for particle detection. However, these instruments are not well-positioned for the analysis of ENDS aerosols when the objective is to assess the particle size distribution in real-time while simultaneously replicating the vapor topography and physiological breathing patterns. Thus, there is a clear need for more advanced and capable technologies for the physical characterization of EC aerosols.

## Engineered non-living robotic systems for ENDS inhalation toxicology

The field of bioinspired engineering has undergone major advancements, offering innovative solutions across various domains, including toxicological assessment. Bioinspired engineering involves designing systems and devices that mimic the structure and function of biological entities to address complex scientific and engineering challenges. This approach has recently gained traction in the evaluation of ENDS[138–141]. A notable development in this field has been the development of the Human Vaping Mimetic Real-Time Particle Analyzer (HUMITIPAA) designed by Kaiser et al. [138]. This innovative system (see below) exemplifies the potential of bioinspired engineering in toxicological research.

### Human vaping mimetic real-time particle analyzer

HUMITIPAA is a biologically inspired, robotic system that integrates laser-based particle detection, the mechanics of respiration, and vaping topography to simulate human vaping behavior. This advanced electro-optical-mechanical system essentially functions as a sensor analogous to one placed at the level of the trachea and is capable of measuring inhaled submicron- and micron-sized particles from EC aerosols in real-time. It operates by mimicking physiological breathing cycles and clinically relevant vaping patterns. The HUMITIPAA categorizes particle size distributions into four ranges—300 nm–1 μm, 1–2.5 μm, 2.5–4 μm, and 4–10 μm—and quantifies the number of particles within each size bracket, as well as providing a cumulative count. Crucially, users can simulate breathing that is clinically evident in both healthy and diseased lung conditions and can adjust vaping topography parameters to reflect different usage patterns. The development of HUMITIPAA was motivated by the sudden emergence of VEA-containing vaping products and the rapid proliferation of e-liquids modified with a wide array of flavorings and other additives, necessitating the need for relevant and practical preclinical tools to evaluate their toxicological profiles. The developers of HUMITIPAA hypothesized that a higher total count of inhaled particles correlates with increased pulmonary toxicity, underscoring the importance of their system in assessing potential health risks associated with vaping products.

Utilizing HUMITIPAA, researchers discovered that even a minimal addition of 1.25% VEA to a 50:50 PG:VG solution resulted in a substantial increase in the number of inhaled particles across all size fractions[138]. Analysis after seven independent vaping sessions, each comprising nine puffs for a total of 63 puffs, revealed an increase in the cumulative particle count from ~38 million to ~58 million particles per cubic centimeter when the e-liquid contained 5% VEA[138]. This incorporation of VEA also shifted the size distribution of the inhaled particles. Specifically, there was a slight decrease in the proportion of submicron particles (from 85.4% to 82.2%), while the proportions of particles sized 1–2.5 μm and 2.5–4 μm increased from 12.7% to 15.5% and from 1.6% to 2.0%, respectively[138]. The researchers postulated that these alterations could in part explain some of the pulmonary pathologies observed in patients with EVALI. Intriguingly, this study also examined the influence of breathing dynamics associated with obstructive (common in asthma and COPD) and restrictive (present in IPF) pulmonary disorders. After correction for volume, the greatest relative increase in the total inhaled particle count was observed with a restrictive breathing pattern (199%), followed by those with obstructive (172%) and healthy (153%) breathing patterns[138]. This finding suggested that the mechanics and patterns of breathing significantly affect the quantity of particles inhaled. Extrapolating from these findings, one may anticipate that vaping a 5% VEA-supplemented e-liquid could result in increased particle deposition in the respiratory tract of patients with IPF, asthma, or COPD compared with individuals with a healthy breathing profile. Additionally, the introduction of nicotine to e-liquids at concentrations ranging from

0.6% to 2.4% slightly reduced the particle count inhaled, although this effect was not statistically significant and was nullified when VEA was also present in the e-liquid[138].

In a subsequent investigation, the same team employed the HUMITIPAA to assess alterations in aerosol particle generation from ECs supplemented with menthol[140]. Their data indicated that adding 0.1% to 8% menthol to a laboratory-prepared 50:50 PG:VG solution significantly elevated the total number of inhaled particles in a dose-dependent manner. Notably, the study revealed that a commercially and highly popular menthol-flavored EC pod produced considerably more 1–10 μm particles than did a control tobacco pod from the same brand with an equivalent nicotine content[140]. Crucially, this study corroborated the in vitro HUMITIPAA results with clinical observations. By examining the COPDGene study, which included the most extensive cohort of EC users at that time, a retrospective analysis discerned an association between the use of menthol-flavored ECs and a decrease in key lung function metrics, specifically the FEV1 to forced vital capacity ratio (FEV1/FVC) and the FEV1 as a percentage of the predicted value[140]. These findings could be pivotal for understanding the respiratory implications of flavored EC use.

The relationships between the size and number of inhaled particles and health risk have been extensively documented in multiple studies[142–146]. For instance, in a systematic review of over 1900 time-series studies, Ren et al. investigated the relationship between the concentration of $PM_{2.5}$ particles and daily respiratory deaths in China[142]. They reported that for every 10 μg/m³ increase in the $PM_{2.5}$ concentration, a 0.30% increase in daily respiratory deaths occured[142]. Furthermore, smaller particles (<2.5 μm) are more likely to penetrate deep into the respiratory tract, causing notable health effects. Mathematical calculations, computational fluid dynamics modeling, and epidemiological studies have demonstrated that ultrafine particles (10–250 nm, also commonly referred to as particles <100 nm) have high deposition rates in the alveolar regions of the lung, which increases their potential for adverse health effects[143–146].

In the context of VEA added to the PG:VG mixture, research indicates that VEA can degrade at the high temperatures used in vaping devices, producing various toxic compounds such as ketones and aldehydes[80,147]. Therefore, one can speculate that the negative health effects of VEA in ECs are due to a combination of both increased particle number and the presence of more toxic chemicals resulting from the degradation of VEA, as well as other harmful agents that may be present in VEA-containing e-liquids, such as THC, flavorings, and nicotine.

## HUMITIPAA's future direction and enhancements

The collective findings above underline the applicability of HUMITIPAA as a bioinspired engineered system that facilitates the relatively rapid assessment (within 1–4 days[139]) of inhaled particle profiles emitted by ECs. Nevertheless, HUMITIPAA requires enhancements in several areas. First, the spectrum of particle size detection must be expanded to include nanoparticles. Despite these technical challenges, it is imperative that advancements in sensor technology be integrated into future versions of HUMITIPAA to capture this critical size range. Second, it is important to establish a clearer association between the biological impact and the increased delivery of particles to pulmonary tissues. Future research should aim to synchronize HUMITIPAA assessments with living biological systems, such as Lung-on-a-Chip. This integration would enable simultaneous acquisition of inhaled particle profiles and exposure of living lung tissue—derived from targeted subject populations—to aerosols from ENDS, thereby facilitating the study of cellular and tissue pathology. Moreover, the incorporation of real-time chemical analysis of EC vapors, in tandem with the physical characterization of the inhaled particle count and size distribution, would greatly enrich the comprehensive evaluation of emerging vaping products. Such advancements would not only refine the precision of toxicological assessments but also bolster the relevance of the findings in the context of public health.

## Broader applicability of MPS and robotic systems

MPS and robotic systems hold substantial potential beyond the study of ENDS. These advanced platforms can be instrumental in assessing the health impacts of poor air quality and other inhalation-based risks across wider populations. For instance, Lung-on-a-Chip models can mimic the human lung microenvironment, providing detailed insights into how different pollutants affect lung biology, cellular responses, and tissue functioning. These models have been utilized to study the toxicological effects of various airborne particles, including ultrafine silica nanoparticles and $PM_{2.5}$ particles[148–151], indicating their potential for the study of industrial pollutants, vehicle emissions, and household air contaminants. Furthermore, these systems can be used to identify protective compounds or measures that could mitigate the adverse effects of inhaled pollutants. By integrating MPS and robotic systems with medium-to-high-throughput screening techniques, researchers can systematically test a wide range of compounds to discover those that offer protective benefits. For example, antioxidant compounds or anti-inflammatory agents can be screened for their efficacy in reducing lung inflammation and oxidative stress caused by pollutant exposure. In high-risk living or working environments, such as industrial settings or urban areas with high pollution levels, MPS and robotic systems can be used to evaluate the effectiveness of protective measures. This includes testing the impact of air purification systems, personal protective equipment, and occupational health interventions. By providing precise and human-relevant data, these systems can inform policy decisions and lead to the implementation of more effective strategies to protect public health. Overall, the broader applicability of MPS and robotic systems extends to various fields, offering valuable insights and solutions to improve public health outcomes related to air quality and inhalation risks. These systems not only enhance our understanding of the toxicological impacts of pollutants but also pave the way for discovery of new protective measures and interventions.

## Summary and outlook

A thorough analysis of the advent of ENDS reveals critical societal and health-related concerns that are instrumental in shaping both future prospects and potential hurdles within this field. ENDS have catalyzed a substantial shift in nicotine consumption patterns, particularly among adolescents and young adults, mandating a comprehensive understanding of their extensive implications.

The documented toxicity of e-liquid constituents such as PG and VG underscores the imperative for the rigorous assessment of ENDS for pulmonary toxicity. Concurrently, the burgeoning variety of flavorings and chemical compounds in e-liquids complicates the evaluation of their health effects. Unfortunately, the extensive product range and the allure of custom-built e-liquids have propelled their widespread adoption. This calls for the progression of research methodologies and toxicological evaluations to parallel the dynamic evolution of ENDS formulations. The EVALI outbreak in 2019 starkly exemplified the repercussions of insufficient scrutiny of ENDS ingredients, underscoring the difficulty in forecasting health outcomes from innovative and unregulated concoctions. Therefore, there is an urgent need for rapid, comprehensive methods to identify potentially harmful constituents prior to consumer exposure.

Currently, a suite of in vitro, ex vivo, in vivo, and in silico techniques are employed by researchers to examine the toxicological effects of vaping products (Fig. 1, Table 1). These include 2D and 3D static cell cultures, such as TWIs, spheroids and organoids, PCLS, animal models, computational models of pulmonary particle deposition, and physicochemical analytical methods. Nevertheless, the utility of preclinical models is hampered by their limited capacity to replicate the intricate 3D multicellularity of lung tissues, vascular flow, and dynamic intercellular interactions, and they typically have a relatively short lifespan. Moreover, these models cannot introduce ENDS aerosols under conditions that mimic rhythmic breathing without disturbing the natural ALI. Additionally, these strategies often lack high-throughput capabilities. The use of animal models is constrained by

considerable interspecies differences that limit extrapolation to humans and ethical concerns. Computational and physicochemical methodologies face challenges related to data quality, model validation, and the limitations inherent in the parameters that can be simulated, as well as the absence of real-time ENDS aerosol generation that emulates actual vaping topography and respiratory mechanics, coupled with a lack of concurrent biological validation.

Emerging Organ-on-Chip technologies and bioinspired robotics, such as the all-primary human Lung Small Airway-on-a-Chip and HUMITI-PAA, show promise in addressing certain limitations of traditional approaches and in facilitating human-relevant modeling of lung patho-physiology and relatively rapid toxicity assessments of ENDS aerosols. Nonetheless, these cutting-edge platforms require further technical and biological refinement as they mature. Their deployment and integration into standard practices have the potential to substantially propel the field of inhalation toxicology forward. These represent forthcoming opportunities that can underpin targeted regulations and public health initiatives aimed at curbing the detrimental effects of ENDS.

Globally, discrepancies in the regulation and public perception of ENDS present major challenges. The absence of universal guidelines and the disparate regulatory approaches among nations foster conditions conducive to public health hazards. Consequently, there is a need for global cooperation to formulate uniform regulatory frameworks. The harmonization of policies and the exchange of best practices could act as safeguards against the dissemination of potentially hazardous ENDS products and ensure a unified effort to protect public health.

In summary, while ENDS may potentially offer harm reduction advantages for adult smokers, their widespread appeal to non-smokers, particularly young people, raises substantial public health issues. The future calls for the invention of novel tools and further improvement of current state-of-the-art bioengineered systems that enable scientists to predict the human-relevant pulmonary toxicity of emerging ENDS products more accurately and rapidly, thereby benefiting society and bolstering the efforts of regulatory bodies. Extensive research, progressive policy-making, and international collaboration are essential for adeptly managing the changing milieu of nicotine delivery mechanisms.

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

## Acknowledgements
This work was supported by the Division of Pulmonary, Allergy, Critical Care, and Sleep Medicine at the University of Pittsburgh and National Institutes of Health grants R01HL159494 and U01EB029085. General: Some of the images in Fig. 1, at least in part, were created using BioRender.com.

## Author contributions
K.H.B. conceived, drafted, and revised the manuscript.

## Competing interests
K.H.B. is a founder and holds equity in Pneumax, LLC.
