## [Peer Review File · Communications Engineering]

This manuscript has been previously reviewed at another Nature Portfolio journal. This document only contains reviewer comments and rebuttal letters for versions considered at *Communications Engineering*.Reviewers' comments:

Reviewer #1 (Remarks to the Author):

The article titled "Living and Non-living Bioinspired Engineered Systems for ENDS Inhalation Toxicology" presented a comprehensive review of current methodologies and the emerging bioengineered systems in the assessment of toxicological impacts of electronic nicotine delivery systems (ENDS). Basically, the essential summarization and induction are necessary to lead the audience to view the whole panorama of the field, but the authors sounded forget the point and were eager to emphasize the evaluation of the ENDS. Secondly, the essential logical catalogs to the evaluation of the toxicity of ENDS is important to make the review readable. Therefore, I do not recommend the publishing of the view in current format. In addition, some comments were listed as the follows.

1. The essential subdirectories are necessary to make the section "Existing Preclinical Living Models for the Pulmonary Toxicological Assessment of ENDS" logic and readable. The same problems were also existing in the late two sections.
2. It is weird that the section of "Computational and Physico-Chemical Analyses of ENDS Liquids and Aerosols" was set a parallel section following "Existing Preclinical Living Models for the Pulmonary Toxicological Assessment of ENDS", since the section "Organ-on-Chip Models of Inhaled Exposure to ENDS Aerosols" looks like the other solid approach to evaluate the toxicity of ENDS instead of the "Computational and Physico-Chemical Analyses of ENDS Liquids and Aerosols", which seems a soft approach to evaluate the toxicity of ENDS.
3. While BEAS-2B cells are extensively used, incorporating data or findings from other relevant cell lines such as A549, NCI-H441, or Calu-3 could provide a more comprehensive view of the toxicological impacts of ENDS. Different cell lines may offer unique insights due to their specific origin, characteristics, and response to toxicants.
4. Is it necessary to consider the title of the review based on the content? since the review included the computational approach to evaluate the toxicity of ENDS, which was beyond the rang of "Bioinspired Engineered Systems".
5. Expanding the discussion to include 3D culture systems, such as spheroids or organoids derived from lung cells, could provide insights into the more physiologically relevant responses to ENDS exposure. These systems better replicate the architecture and function of lung tissue compared to traditional 2D cultures.
6. To enhance the comprehensiveness and depth of the review on toxicological impacts of ENDS exposure, it is recommended valuable insights into the cellular and molecular changes by ENDS exposure. For example, gene expression profiling, signaling pathway analysis, proteomics and metabolomics and some other biological information could be considered.
7. For the various systems reviewed in this review, it is advisable to consider organizing the information into a table format, offering a clearer and more concise overview of each system. The table could be structured as follows:
 - a) System classification: traditional in vitro models, advanced engineered systems (e.g., organ-on-a-chip), bioinformatics tools, animal models...
 - b) Components: Describe the main components or features of each system. For engineered systems, this might include the type of cells used, the materials for the chip, and the presence of microfluidic channels. For bioinformatics tools, list the main software or algorithms employed.
 - c) Characteristics (pros and cons): outline the key advantages and limitations of each system.
 - d) Provided biological information
 - e) Application examples

Reviewer #2 (Remarks to the Author):

Dr. Benam reviews interesting topics regarding new electronic smoking techniques, defined as electronic nicotine delivery systems (ENDS). As mentioned, there have been several reports in the last several years regarding serious side effects, illness or death after the use of such devices, linked to certain compounds added into smoking liquids. Therefore, epidemiological studies and methods to study ENDS toxicity are relevant for public health regarding these new smoking options in the current day. As described by the reviewer, there are potentially thousands of

variations in ENDS compound compositions, providing a need for rapid screening methods to identify compounds that are health hazards. One example presented is the additive Vitamin E Acetate, used before safely for instance in topical applications, but a likely cause of serious toxic effects when used in certain ENDS. It is also interesting to discuss the discrepancies in policies around the world and how they affect public health. The article provides a complete overview of current methods used in ENDS toxicology and some related fields. The article could be improved by providing a clearer overview of the currently known epidemiological data, better description and summary quantifications/statistics of old, current and newly developing models in the discovery of ENDS toxicology, the use of organ-on-a-chip methods in studying toxicology or drug discovery in the study of ENDS-related smoking, and in the health field in general. The comments below may aid in addressing these improvements and make the article more relevant to ENDS studies and the wider field of integrative models and/or lung-based disorders.

1. In the introduction, some statistics are presented about the use of ENDS and (e-)cigarettes. The relevance of this for public health could be improved by providing additional data on negative health effects, e.g. what percentage of ENDS users experience negative health effects and what is the severity, based on some of the currently known studies. What was the range of effects and their prevalence for instance in the studies of Vitamin E Acetate use in ENDS? Besides Vitamin E Acetate, are there other known compounds with similar effects, and perhaps the overall number of compounds currently being investigated for health hazards in ENDS use? This would give a better understanding of whether the main additives PG/VG should already be avoided, or if other additives provide greater health risk.

2. Describing current models for ENDS toxicology using cell-based systems, the author provides a useful overview of various biological models and which components are modeled. Some use only lung cells, whereas others aim to be more integrative by adding vasculature or other physiological model aspects, and animal models (complete system) are also explored. Based on what is known about ENDS toxicology from these studies, is it known what would be the most important parts to model related to observed effects? Are effects primarily happening in the lungs, or is there significant evidence of multiple systems involved in reactions to potentially toxic compounds via use of ENDS? This would help direct the field in which can of models to prioritize. I understand this could potentially be different for each individual compound, but I think it would still be relevant to have some better insight in this.

3. In the paragraph about in silico and physical/chemical screening methods, it would be useful to provide benchmark numbers on how successful the current state-of-the-art methods are in identifying or predicting toxic compounds related to ENDS, or their deposition dynamics as studied in some of the references. One particular study is cited in the manuscript (Zarini et al.) that screened over 200 compounds using the QSAR model. Was any benchmarking used in this study to assess the strength of their model by using known toxic compounds? The author remarks there are limitations using these kinds of methods, but without any quantification of those limitations it is hard to understand how these methods stand against other methods presented in the manuscript.

4. Organ-on-chip models, also referred to as microphysiological systems (MPS), provide a relevant model to study physiological systems in a scalable manner. Therefore, the use of MPS appears like a good option in screening compounds used in ENDS as suggested by the author. As an example a study from the author regarding COPD is presented, which provided valuable new insight in COPD and novel biomarkers. Are there other fields in which MPS successfully helped in identifying toxic compounds or new therapeutic targets?

5. In part of the review, smoking patterns and inhalation of particles are discussed, where it is observed that inhaled particle size and number is increased when Vitamin E Acetate is added to the PG/VG mix. From these studies, and in general, is it known how particle size and number related to health risk? Is the health risk in this particular example purely from Vitamin E Acetate inhalation, or are particular particles the main risk?

6. The MPS and Robotic systems described potentially have a very relevant wider use beyond the study of ENDS. One can imagine the wide population effect of poor air quality and other inhalation/lung based health risks could be studied using this method. It could be briefly

commented how these methods are applicable beyond the current use. Also, could the MPS and Robotic systems be used to identify protective compounds or other protective measures to ameliorate the effects of inhaled pollutants or particles in high risk living or working environments?

Reviewer #3 (Remarks to the Author):

Reviewer #1:

I thank this Reviewer for stating our manuscript presents “a comprehensive review of current methodologies and the emerging bioengineered systems in the assessment of toxicological impacts of electronic nicotine delivery systems (ENDS)” and address their specific comments as detailed below.

- 1. *The essential summarization and induction are necessary to lead the audience to view the whole panorama of the field, but the authors sounded forget the point and were eager to emphasize the evaluation of the ENDS.***

I appreciate the reviewer’s feedback, and have revised the “Societal and Health Impacts of Emerging ENDS” section to incorporate a broad discussion on the negative health effects of ENDS, focusing primarily on the pulmonary and cardiovascular systems, which are most relevant to the scope of this manuscript. Additionally, I have included detailed statistics on the percentage of mortality and hospitalization associated with EVALI, as well as lung toxicity associated with diacetyl, a compound generally considered safe as a food and drink additive. These revisions aim to enhance the reader's understanding of the broader context and the specific health risks posed by ENDS.

- 2. *The essential subdirectories are necessary to make the section “Existing Preclinical Living Models for the Pulmonary Toxicological Assessment of ENDS” logic and readable. The same problems were also existing in the late two sections.***

I have added subdirectories to the requested section and other relevant parts of the manuscript to ensure a clearer and more structured presentation. These changes aim to improve the overall coherence and ease of navigation through the manuscript, addressing the reviewer's concern effectively. I believe these revisions will significantly enhance the manuscript's readability and logical flow.

- 3. *It is weird that the section of “Computational and Physico-Chemical Analyses of ENDS Liquids and Aerosols” was set a parallel section following “Existing Preclinical Living Models for the Pulmonary Toxicological Assessment of ENDS”, since the section “Organ-on-Chip Models of Inhaled Exposure to ENDS Aerosols” looks like the other solid approach to evaluate the toxicity of ENDS instead of the “Computational and Physico-Chemical Analyses of ENDS Liquids and Aerosols”, which seems a soft approach to evaluate the toxicity of ENDS.***

I understand this Reviewer’s concern regarding the sequence of sections in the manuscript. To address this, I have reorganized the sections to place "Organ-on-Chip Models of Inhaled Exposure to ENDS Aerosols" immediately after the "Existing Preclinical Living Models for the Pulmonary Toxicological Assessment of ENDS" section. This reorganization aligns with the Reviewer’s suggestion and enhances the logical flow of the manuscript.

- 4. *While -2B cells are extensively used, incorporating data or findings from other relevant cell lines such as A549, NCI-H441, or Calu-3 could provide a more comprehensive view of the toxicological impacts of ENDS. Different cell lines may offer unique insights due to their specific origin, characteristics, and response to toxicants.***

I appreciate the reviewer's comment. While I agree that different cell lines can provide unique insights due to their specific origins and characteristics, as stated in the manuscript, cell lines are either immortalized or cancerous. This often leads to biological and functional differences from what is naturally observed in the human body. For instance, Beas-2B cells (an immortalized line), as stated in

the manuscript, cannot differentiate to form a mucosal mucociliated airway epithelium, which is critical for studying respiratory airway toxicity. Similarly, A549 cells are derived from lung adenocarcinoma and are hypotriploid with a modal chromosome number of 66, observed in about > 20% of the cell population (DOI: 10.1165/rcmb.2021-0048ED; see ATCC page for karyotype details). They are a model of alveolar type II epithelial cells and have been widely used in lung cancer research as well as studies of alveolar type II pneumocytes. However, A549 cells have notable limitations. They do not express surfactant proteins, which are crucial for the normal function of type II alveolar cells in reducing surface tension in the lungs. Additionally, A549 cells exhibit altered metabolic and signaling pathways due to their cancerous nature, which can lead to responses that differ significantly from those of normal, non-cancerous alveolar cells. Moreover, they lack the ability to differentiate into *in vivo*-like type I alveolar cells, further constraining their relevance for ENDS toxicity assessment at the level of alveoli. Importantly, the use of cell lines does not capture inter-individual variability across the population. Hence, under the “**2D and 3D Static Cell Culture Models**” section, I highlighted the primary human airway epithelial cells (hAEPs) for their ability to provide a more physiologically relevant model system. Primary hAEPs can be derived from different donors, providing a more accurate representation of the genetic and phenotypic diversity found in the human population. This diversity is fundamental for understanding how different individuals might respond to ENDS exposure. I hope this addresses the Reviewer’s concern and clarifies our approach.

5. ***Is it necessary to consider the title of the review based on the content? since the review included the computational approach to evaluate the toxicity of ENDS, which was beyond the rang of “Bioinspired Engineered Systems”.***

I have updated the title to "**Multidisciplinary Approaches in ENDS Toxicology: Emergence of Living and Non-living Bioinspired Engineered Systems**". This revised title captures the diversity of preclinical approaches discussed in our review, including 2D/3D cell cultures, PCLS, animal models, and *in silico* approaches. At the same time, it retains the key message of the paper, which highlights the emergence and significance of Organs-on-Chips and robotic systems in ENDS toxicology.

6. ***Expanding the discussion to include 3D culture systems, such as spheroids or organoids derived from lung cells, could provide insights into the more physiologically relevant responses to ENDS exposure. These systems better replicate the architecture and function of lung tissue compared to traditional 2D cultures.***

I have now revised the manuscript to include a subsection on lung spheroids and organoids under the "**Existing Preclinical Living Models for the Pulmonary Toxicological Assessment of ENDS**" section.

7. ***To enhance the comprehensiveness and depth of the review on toxicological impacts of ENDS exposure, it is recommended valuable insights into the cellular and molecular changes by ENDS exposure. For example, gene expression profiling, signaling pathway analysis, proteomics and metabolomics and some other biological information could be considered.***

I appreciate the reviewer's recommendation. While this is a valuable area of research, it is beyond the primary focus of this manuscript. The aim of our review is to provide a comprehensive overview of the existing and emerging preclinical models used to study the inhalation toxicology of ENDS. This includes traditional 2D and 3D cell cultures, organoids and spheroids, precision-cut lung slices (PCLS), animal models, *in silico* approaches, physico-chemical analyses, and advanced bioengineered systems like Organs-on-Chips and robotic platforms. As relevant, I have mentioned specific pathologies and cellular responses observed in different model systems. For example, I discussed inflammation, impaired lung function, and increased respiratory diseases observed in various studies. However, an in-

depth exploration of gene expression profiling, signaling pathway analysis, proteomics, and metabolomics is outside the scope of this review. Our primary goal is to highlight the capabilities and limitations of the current models used for evaluating the toxicological impacts of ENDS, emphasizing the importance of these models in advancing our understanding and regulatory approaches to ENDS products. I believe that maintaining this focus allows us to provide a more targeted and informative review for researchers and policymakers interested in the methodologies used to assess the health risks associated with ENDS exposure.

- 8. For the various systems reviewed in this review, it is advisable to consider organizing the information into a table format, offering a clearer and more concise overview of each system. The table could be structured as follows:**
- a. System classification: traditional in vitro models, advanced engineered systems (e.g., organ-on-a-chip), bioinformatics tools, animal models.**
 - b. Components: Describe the main components or features of each system. For engineered systems, this might include the type of cells used, the materials for the chip, and the presence of microfluidic channels. For bioinformatics tools, list the main software or algorithms employed.**
 - c. Characteristics (pros and cons): outline the key advantages and limitations of each system.**
 - d. Provided biological information.**
 - e. Application examples.**

In response to the Reviewer's comment, I have now generated a comprehensive table (**Table 1**). This table includes the classification of systems, main components, characteristics (pros and cons), provided biological information, and application examples. The inclusion of this table enhances the clarity and readability of the manuscript by offering a structured summary of the different approaches used in the toxicological assessment of ENDS.

Reviewer #2:

I thank this Reviewer for stating that “*the article provides a complete overview of current methods used in ENDS toxicology and some related fields*”. I address their specific comments as detailed below.

- In the introduction, some statistics are presented about the use of ENDS and (e-)cigarettes. The relevance of this for public health could be improved by providing additional data on negative health effects, e.g. what percentage of ENDS users experience negative health effects and what is the severity, based on some of the currently known studies. What was the range of effects and their prevalence for instance in the studies of Vitamin E Acetate use in ENDS? Besides Vitamin E Acetate, are there other known compounds with similar effects, and perhaps the overall number of compounds currently being investigated for health hazards in ENDS use? This would give a better understanding of whether the main additives PG/VG should already be avoided, or if other additives provide greater health risk.***

In response to this Reviewer’s suggestions, I have revised our manuscript to include detailed information on the negative health effects associated with the use of ENDS. I have also provided data on the prevalence and severity of these effects, particularly focusing on conditions such as EVALI and the impact of specific compounds like VEA and diacetyl.

Recent research has highlighted a range of adverse health effects linked to ENDS use. For example, an analysis of surveys and electronic health records from over 175,000 participants in the *All of Us* Research Program led by the NIH found that ENDS users were 19% more likely to develop heart failure compared to non-users. Additionally, chronic ENDS use has been shown to impair vascular function, as evidenced by decreased flow-mediated dilation and increased levels of inflammatory markers and endothelial adhesion molecules in users.

In the pulmonary medicine space, studies have consistently shown that ENDS use increases the incidence of respiratory symptoms among adolescents and young adults. Research has found increased reporting of wheezing, bronchitic symptoms, and shortness of breath among current and past ENDS users. These findings underscore the considerable respiratory risks associated with ENDS use and emphasize the need for increased awareness and regulatory measures to mitigate these health impacts.

Moreover, the EVALI epidemic between 2019 and 2020, which resulted in 2,807 reported cases and 68 deaths in the United States, illustrates the severe health consequences of certain compounds in ENDS liquids. Vitamin E acetate was identified as a primary culprit, linked to a 2.4% mortality rate among affected individuals. The majority of EVALI patients required hospitalization, with a substantial proportion needing intensive care and intubation. This underscores the importance of monitoring and regulating additives in ENDS products.

I also highlighted the risks associated with diacetyl, a compound used to impart a buttery flavor to vaping liquids. Despite being considered safe for ingestion, diacetyl has been linked to severe respiratory conditions such as bronchiolitis obliterans and COPD when inhaled. This compound was particularly prevalent in EC vapors, raising concerns about its safety and the need for stringent regulation to protect public health.

By incorporating these detailed findings, I aimed to provide a more comprehensive understanding of the negative health effects associated with ENDS use and the specific risks posed by various compounds. I believe this enhances the relevance of our manuscript for public health considerations and addresses the points raised in this Reviewer.

2. ***Describing current models for ENDS toxicology using cell-based systems, the author provides a useful overview of various biological models and which components are modeled. Some use only lung cells, whereas others aim to be more integrative by adding vasculature or other physiological model aspects, and animal models (complete system) are also explored. Based on what is known about ENDS toxicology from these studies, is it known what would be the most important parts to model related to observed effects? Are effects primarily happening in the lungs, or is there significant evidence of multiple systems involved in reactions to potentially toxic compounds via use of ENDS? This would help direct the field in which can of models to prioritize. I understand this could potentially be different for each individual compound, but I think it would still be relevant to have some better insight in this.***

The impact of vaping has been predominantly studied using models of the lung due to the inhaled exposure to EC vapors. These studies have provided significant insights into the respiratory effects of vaping, including inflammation, impaired lung function, and increased risk of respiratory diseases. However, there are also preclinical and clinical reports that have examined the effects of vaping on the cardiovascular and immune systems, indicating potential systemic health risks. Despite these important findings, the focus of this paper is confined to the pulmonary impacts of vaping, and therefore, the cardiovascular and immune system effects are outside the scope of this discussion and are not addressed here. I have revised the manuscript to add this text and added appropriate citations.

In response to the reviewer's question about "*what would be the most important parts to model related to observed effects*", it is critical to highlight the diversity and complexity of models used to study the impact of ENDS (all discussed under sections of our manuscript). The majority of investigators have focused on lung models, primarily targeting the lung epithelium, which is the first site of major exposure to inhaled EC aerosols. These models range from simpler *in vitro* systems using isolated lung epithelial cells (line or primary) to more complex models like PCLS and Organs-on-Chips. Simpler models, while valuable for understanding basic cellular responses such as inflammation and cytotoxicity, are limited in their ability to provide comprehensive insights into tissue-level interactions and the broader physiological impact. In contrast, more complex models offer a higher level of physiological relevance of the lung tissue. Therefore, while simpler models are useful for initial screenings and mechanistic studies, more complex and integrative models are essential for capturing broader biological effects of exposure to ENDS vapors.

3. ***In the paragraph about in silico and physical/chemical screening methods, it would be useful to provide benchmark numbers on how successful the current state-of-the-art methods are in identifying or predicting toxic compounds related to ENDS, or their deposition dynamics as studied in some of the references. One particular study is cited in the manuscript (Zarini et al.) that screened over 200 compounds using the QSAR model. Was any benchmarking used in this study to assess the strength of their model by using known toxic compounds? The author remarks there are limitations using these kinds of methods, but without any quantification of those limitations it is hard to understand how these methods stand against other methods presented in the manuscript.***

Zarini et al. (DOI: 10.1021/acs.chemrestox.0c00136) investigated the applicability of in silico methods for the initial safety assessment of ENDS. The authors evaluated the potential of QSAR models to predict the oral and inhalation toxicity of e-liquid constituents. The study involved screening over 200 compounds using QSAR models. The authors applied LC50 values (lethal concentration for 50% of the population) to assess the inhalation toxicity of e-liquid ingredients. Specifically, they evaluated 28 e-liquid ingredients for rats and 23 for mice. Among these, only 22 and 19 compounds served as training data sets for rats and mice, respectively. The inhalation toxic potential was studied for only 10 compounds (6 for rats and 4 for mice). While the study provides valuable insights (predictability of 86-88%), there were major limitations to the model used. (1) The primary data for model training and benchmarking were derived from rodent (not human) studies. This poses a

challenge as the toxicological responses in rodents may not accurately reflect those in humans, potentially leading to less reliable predictions for human health risks. (2) The utilized models relied heavily on LC50 data, which classifies doses based on their lethality potential. This approach necessitates the use of high-dose exposures to determine lethal concentrations, which may not be representative of typical human exposures to e-liquid ingredients. Additionally, LC50 data are often limited and may not be available for all relevant compounds, restricting the model's applicability and accuracy. Therefore, this study demonstrates that while in silico methods such as QSAR models can be useful for the initial screening of e-liquid ingredients' toxicity, there are major limitations that need to be addressed. The reliance on rodent data and the necessity for LC50 values highlight the need for more human-relevant data and alternative approaches to improve the accuracy and reliability of these models for ENDS safety assessments.

For other in silico studies cited in our manuscript, no benchmarking was provided. For instance, the study by Haghnegahdar et al. (DOI: 10.1080/02786826.2018.1447644) generated valuable insights into the effects of anatomical variability on particle deposition; however, lack of explicit benchmarking with human data sets or known toxic compounds means that the predictive accuracy of the model may not be fully validated. Similarly, no benchmarking was performed by Zhao et al. (DOI: 10.1016/j.combiomed.2021.104333) study.

I have revised the manuscript to clarify and include these key points.

- 4. Organ-on-chip models, also referred to as microphysiological systems (MPS), provide a relevant model to study physiological systems in a scalable manner. Therefore, the use of MPS appears like a good option in screening compounds used in ENDS as suggested by the author. As an example, a study from the author regarding COPD is presented, which provided valuable new insight in COPD and novel biomarkers. Are there other fields in which MPS successfully helped in identifying toxic compounds or new therapeutic targets?**

Yes, microphysiological systems have successfully been applied in several fields beyond ENDS toxicology. In drug development and safety testing, Liver-on-a-Chip models predict human hepatotoxicity and detect cross-species toxicological risks (DOI: 10.1126/scitranslmed.aax5516; DOI: 10.1038/s43856-022-00209-1). In cancer research, Tumor-on-a-Chip models replicate tumor microenvironments, aiding in the discovery of anti-cancer drugs (DOI: 10.1038/s42003-023-05531-5; DOI: 10.1016/j.celrep.2017.09.043; DOI: 10.1002/adhm.202201555; DOI: 10.1186/s12951-024-02625-y; DOI: 10.1016/j.bioadv.2024.213915). Neurological diseases have benefited from Brain-on-a-Chip and Blood Brain Barrier-on-a-Chip models, facilitating studies on neurodegenerative diseases like Alzheimer's and Parkinson's, leading to new drug targets and therapeutic candidates (DOI: 10.1039/c4lc00962b; DOI: 10.1074/jbc.RA120.013325; DOI: 10.3390/biom12081136; DOI: 10.1038/s41467-019-10588-0; DOI: 10.1038/s41467-021-26066-5; DOI: 10.1007/s00018-021-04047-7). Additionally, cardiovascular research has utilized Heart-on-a-Chip models to study cardiac physiology, identify cardiotoxic compounds, and ensure safer therapeutic options (DOI: 10.1039/c5lc01356a; DOI: 10.1016/j.biomaterials.2018.07.062; DOI: 10.1126/scitranslmed.aax8005; DOI: 10.1039/c7lc00740j). I have revised the manuscript to add this text with appropriate citations.

- 5. In part of the review, smoking patterns and inhalation of particles are discussed, where it is observed that inhaled particle size and number is increased when Vitamin E Acetate is added to the PG/VG mix. From these studies, and in general, is it known how particle size and number related to health risk? Is the health risk in this particular example purely from Vitamin E Acetate inhalation, or are particular particles the main risk?**

The relationship between the size and number of inhaled particles and health risk has been extensively documented in multiple studies. For instance, in a systematic review of over 1,900 time-

series studies, Ren et al. investigated the relationship between the concentration of PM_{2.5} particles (defined as the number of particles per unit volume) and daily respiratory deaths in China. They found that for every 10 µg/m³ increase in PM_{2.5} concentration, there was a 0.30% increase in daily respiratory deaths (DOI: 10.1155/2017/5806185). Furthermore, smaller particles (<2.5 µm) are more likely to penetrate deep into the respiratory tract, reaching the alveoli and causing significant health effects. Mathematical calculations, computational fluid dynamics modeling, and epidemiological studies have demonstrated that ultrafine particles (10–250 nm, also commonly referred to particles <100 nm) have high deposition rates in the alveolar regions of the lung, which increases their potential for adverse health effects (DOI: 10.4209/aaqr.2020.01.0033; DOI: 10.4209/aaqr.2020.02.0067; DOI: 10.1007/s11356-023-31248-3; DOI: 10.1007/s00038-019-01202-7).

In the context of Vitamin E Acetate (VEA) added to the PG/VG mixture, research indicates that VEA can degrade at the high temperatures used in vaping devices, producing various toxic compounds such as ketones and aldehydes (DOI: /10.3389/fpubh.2021.765168; DOI: 10.1056/NEJMc2000231). Therefore, I 'speculate' that the negative health effects of VEA in ECs are due to a combination of both increased particle number and the presence of more toxic chemicals resulting from the degradation of VEA, as well as other harmful agents that may be present in VEA-containing e-liquids, such as THC, flavorings, and nicotine.

I have revised the manuscript to include these points.

6. *The MPS and Robotic systems described potentially have a very relevant wider use beyond the study of ENDS. One can imagine the wide population effect of poor air quality and other inhalation/lung-based health risks could be studied using this method. It could be briefly commented how these methods are applicable beyond the current use. Also, could the MPS and Robotic systems be used to identify protective compounds or other protective measures to ameliorate the effects of inhaled pollutants or particles in high risk living or working environments?*

This is a very important and insightful comment. Microphysiological systems and Robotic systems hold substantial potential beyond the study of ENDS. These advanced platforms can be instrumental in assessing the health impacts of poor air quality and other inhalation-based risks across wider populations. For instance, Lung-on-a-Chip models can mimic the human lung microenvironment, providing detailed insights into how different pollutants affect lung biology, cellular responses, and tissue functioning. Such models have been utilized to study the toxicological effects of various airborne particles, including ultrafine silica nanoparticles and PM_{2.5} particles (DOI: 10.1126/science.1188302; DOI: 10.1021/acssensors.8b01672; DOI: 10.1021/acsbio.0c00221; DOI: 10.1016/j.ecoenv.2021.112601), indicating their potential for study of industrial pollutants, vehicle emissions, and household air contaminants.

Furthermore, these systems can be used to identify protective compounds or measures that could mitigate the adverse effects of inhaled pollutants. By integrating MPS and Robotic systems with medium-to-high-throughput screening techniques, researchers can systematically test a wide range of compounds to discover those that offer protective benefits. For example, antioxidant compounds or anti-inflammatory agents can be screened for their efficacy in reducing lung inflammation and oxidative stress caused by pollutant exposure.

In high-risk living or working environments, such as industrial settings or urban areas with high pollution levels, MPS and Robotic systems can evaluate the effectiveness of protective measures. This includes testing the impact of air purification systems, personal protective equipment, and occupational health interventions. By providing precise and human-relevant data, these systems can inform policy decisions and lead to the implementation of more effective strategies to protect public health.

Overall, the broader applicability of MPS and Robotic systems extends to various fields, offering valuable insights and solutions to improve public health outcomes related to air quality and inhalation risks. These systems not only enhance our understanding of the toxicological impacts of pollutants but also pave the way for discovering new protective measures and interventions.

I have revised the manuscript to include these points.

REVIEWERS' COMMENTS:

Reviewer #1 (Remarks to the Author):

The manuscript titled "Multidisciplinary approaches in ENDS toxicology: emergence of living and non-living bioinspired engineered systems" already fixed all the issues I listed. The quality of the manuscript improved a lot, therefore I recommend the editor consider its publishing after the minor modification of language.

Reviewer #2 (Remarks to the Author):

The revised manuscript by Dr. Benam has shown significant improvement, providing more comprehensive details, including biological quantification and epidemiological statistics. The subsectioning of topics has been enhanced, and the manuscript now mentions applications beyond ENDS studies. All the comments I previously provided have been addressed in the rebuttal and incorporated into the revised manuscript. It is explicitly stated in the revised manuscript that the models for ENDS discussed in the review are focused exclusively on pulmonary/lung biology, although effects on other parts of the organ system are acknowledged, as per my comment 2. Therefore, I find that the review comments have been satisfactorily addressed.

Reviewer #3 (Remarks to the Author):

I co-reviewed this manuscript with one of the reviewers who provided the listed reports. This is part of the Communications Engineering initiative to facilitate training in peer review and to provide appropriate recognition for Early Career Researchers who co-review manuscripts.

Reviewer #1:

I thank this Reviewer for their positive feedback and appreciate their recommendation for publication.

- 1. The manuscript titled “Multidisciplinary approaches in ENDS toxicology: emergence of living and non-living bioinspired engineered systems” already fixed all the issues I listed. The quality of the manuscript improved a lot, therefore I recommend the editor consider its publishing after the minor modification of language.***

The manuscript has been revised and necessary minor modifications have been made to enhance its readability.

Reviewer #2:

I thank this Reviewer for their satisfaction with the improvements and stating that the comments “have been satisfactorily addressed”.

Reviewer #3:

I appreciate the time and efforts of this Reviewer in evaluating the manuscript.